An automated information extraction system from the knowledge graph based annual financial reports

http://orcid.org/0000-0003-3527-6544 Mohsin Syed Farhan 1 sp17phcs0003@maju.edu.pk
http://orcid.org/0000-0002-9490-7943 Jami Syed Imran 1
http://orcid.org/0000-0003-3660-065X Wasi Shaukat 1
http://orcid.org/0000-0002-5656-0416 Siddiqui Muhammad Shoaib 2
1 Department of Computer Science, Muhammad Ali Jinnah University , Karachi, Sindh , Pakistan
2 Faculty of Computer and Information Systems, Islamic University of Madinah , Madinah , Saudi Arabia
Shuja Junaid
Electronic publication date: 2024 May 13
Publication date: 2024
Volume: 10
Electronic Location ID: e2004
Received 2023 Dec 29; Accepted 2024 Apr 1
Copyright: © 2024 Mohsin et al.
Copyright year: 2024
Copyright holder: Mohsin et al.
License: This is an open access article distributed under the terms of the Creative Commons Attribution License, which permits unrestricted use, distribution, reproduction and adaptation in any medium and for any purpose provided that it is properly attributed. For attribution, the original author(s), title, publication source (PeerJ Computer Science) and either DOI or URL of the article must be cited.
License URL: https://creativecommons.org/licenses/by/4.0/

Keywords: Ontology, Information extraction, Annual financial reports, Knowledge graph, Semantic web, Artificial intelligence

Funding: Islamic University of Madinah, KSA Muhammad Ali Jinnah University This work was supported by Islamic University of Madinah, KSA and Muhammad Ali Jinnah University, Karachi, Pakistan. The funders had no role in study design, data collection and analysis, decision to publish, or preparation of the manuscript.

==============================
This article presents a semantic web-based solution for extracting the relevant information automatically from the annual financial reports of the banks/financial institutions and presenting this information in a queryable form through a knowledge graph. The information in these reports is significantly desired by various stakeholders for making key investment decisions. However, this information is available in an unstructured format making it much more complex and challenging to understand and query manually or even through digital systems. Another challenge that makes the understanding of information more complex is the variation of terminologies among financial reports of different banks or financial institutions. The solution presented in this article signifies an ontological approach to solving the standardization problems of the terminologies in this domain. It further addresses the issue of semantic differences to extract relevant data sharing common semantics. Such semantics are then incorporated by implementing their representation as a Knowledge Graph to make the information understandable and queryable. Our results highlight the usage of Knowledge Graph in search engines, recommender systems and question-answering (Q-A) systems. This financial knowledge graph can also be used to serve the task of financial storytelling. The proposed solution is implemented and tested on the datasets of various banks and the results are presented through answers to competency questions evaluated on precision and recall measures.

Introduction

Annual financial reports are presented by companies and institutions at the end of every financial year. These reports depict the yearly financial positions of the companies and their credibility in terms of profit or loss in the market. The major components of annual financial reports, as presented in Stainbank & Peebles (2006) are shown in Fig. 1.

Figure 1 Different components of annual report (Source: https://www.wallstreetmojo.com/annual-report/).

The content and format of these reports vary according to the various accounting standards and the individual policies of the banks. Due to the lack of a common structure, investors find it difficult to understand the report. For clarity, various consultants consider annual financial reports as a key component in evaluating the financial health of the organization. Therefore, this work targets only the statement of financial position which is mostly composed of data in tabular form. These annual statements of various banks serve as a dataset that is available in textual format and thus requires preprocessing.

Background & motivation

The potential use cases for this research are explained below.

Alice receives her pension monthly. She is interested in investing it to gain the highest profit/earnings after tax. For this purpose, she has gained access to the publicly available financial reports of various banks. She is trying to find her query by looking at the values and terminologies in the annual financial reports.

Bob is interested in investing in the stock shares of the banks that offer the best cash dividend per share value. To get these insights, he is trying to search for different values from the annual reports.

Carol needs a loan on suitable terms to invest in her business. She is working with various financial reports of banks to search for the banks that give loans at low markup.

In these use cases, the customers are coming across with following challenges: The reports are mostly lengthy, detailed, and available in unstructured format and most of the contents may not be relevant for investors.

The financial reports are in a human-readable form. Machines cannot understand these reports therefore, they are not queryable in an automated way.

Each company/institution uses its terminologies due to the lack of standardization. This leads to a huge amount of semantic differences.

The current trends suggest using various recommendations through blogs, reviews, and posts on social media. There are two issues with such options: i) They are based on user experience which may or may not be the true reflection of actual data.

ii) They may not contain relevant information as desired by the user.

Objectives

In order to resolve the issues listed above, an ontology-based information extraction (OBIE) is proposed in which the information extraction process heavily relies on ontologies and has the potential to assist the development of the semantic web (Wimalasuriya & Dou, 2010).

For adapting the system for users as mentioned in Use Cases, the competency questions are designed to show how the system will be used. The competency questions guide the development of the ontology to ensure that it addresses all the inquiries outlined within them. These competency questions are discussed in later subsections. We have used the ontology for matching with the terminologies available in the annual financial reports along with the relevant values of those terminologies. We have also used a regular expression to filter out the precise terminologies to get the relevant values of each terminology.

These reports are in the textual format, containing the financial information about the companies in the form of key terminologies. A Python-based web crawler is developed which is used to crawl the company’s website to search and retrieve the annual financial reports. The details are discussed in later subsections.

The annual financial report in a textual format is converted into the W3C-compliant CSV format by mapping with the ontology to extract the relevant data which helps in standardization. The relevant data is then transformed into the financial knowledge graph (Zehra et al., 2021) to represent it in a machine-readable format for querying competency questions. This research work is the extension of Zehra et al., (2021) with the following contributions: i) An automated information extraction system has been proposed.

ii) The system has been applied to various banks to generate precision and recall values to effectively answer the competency questions.

iii) The existing ontology in Zehra et al. (2021) has been extended with 3,849 triples and 730 concepts.

iv) Transformation of extracted data to the knowledge graph.

In the proposed information extraction system, we attempted to achieve the maximum amount of precision in results along with the higher recall value. The generated results show that our system provides precise information extraction results while extracting the data from the annual financial reports by matching with the ontology. The details are mentioned in the results section.

The rest of the article is organized as follows. The literature review is available in the next section, followed by the methodology section which contains information regarding information extraction, competency questions, semantic differences, and different steps involved in the entire process, the results of the competency questions and queries are described in the results section, while the article ends with a conclusion, future work, and references.

Related work

The last few years have witnessed an increase in the number of research articles in the area of information extraction. Figure 2 shows the number of articles published in renowned journals, the sources include Google Scholar (https://scholar.google.com/), IEEE (https://www.ieee.org/), ACM (https://www.acm.org/), Springer (https://link.springer.com/) and ScienceDirect (https://www.sciencedirect.com/). However, this figure further shows that ontology-based information extraction articles (shown as a line graph) do not get much attention in domain-specific extraction of information. The research article selection criteria encompass articles published from 2018 onwards.

Figure 2 Year-wise graphical representation of the published research article.

The following subsections detail the methodologies used in prior works from the context of information gathering and information extraction. The comparative analysis along with the comparison with our approach are presented in subsequent sections.

Information gathering

The work in Sowunmi et al. (2018) proposed a framework, based on the semantic web, learners’ profiles, resources ontology, and a semantic web search engine. This framework is divided into two components: the general crawler and the RDF crawler, both of which are utilized for personalized information retrieval of educational content from the Internet. The search results are filtered based on semantics and user preferences using a reasoning inference engine. A ranking algorithm is employed to assign weights and rank the search results. The system and search management are accessed through a user interface.

The idea of representing the Hundred Schools of Thought by using a knowledge graph is presented by Wei & Liu (2019). Various tools and techniques are used to accomplish this task. Initially, an ontology generation process is initiated to specify the knowledge graph’s domain and relationships, which results in a better systematic and arranged presentation of the information on the Hundred Schools of Thought. A web crawler is used to collect the relevant data from the websites and utilize it to populate the knowledge graph. To ensure a complete database of knowledge about the Hundred Schools of Thought, the web crawler, after a certain period, scans and gathers data from certain websites. A graph database Neo4j is used for managing and storing knowledge graphs. Neo4j can effectively store, retrieve, and query graph data making it simple to navigate and explore the relationships between the schools, philosophers, and their beliefs. The proposed approach combines ontology development, web crawling, and the use of Neo4j as a knowledge graph database to build a comprehensive representation of the Hundred Schools of Thought. This knowledge graph is a helpful tool for understanding and learning about the conventional doctrines and philosophies of many schools.

An ontological method is presented by Hong, Lee & Yu (2019), for automated information gathering on flooring, ceiling materials, and interior wall finishing materials from four different product producers. The suggested method is using a web crawler to browse and gather pertinent data from construction websites that deal with building supplies. The classification of construction and material information is then particularly addressed by ontology. Data about construction materials may be collected and categorized automatically thanks to web crawling and ontology development.

An ontology-based web crawler (WCO) is introduced by Ibrahim & Yang (2019), for retrieving relevant information in the field of education. The WCO utilizes a crawler to gather information by evaluating the relevance between a user’s query phrases and facts within a specific domain’s reference ontology. To rank the crawled concept phrases, a similarity ranking method is employed. This approach enables the discovery of desired pages that may be overlooked by traditional web crawlers focused on educational content. The process begins with the construction of an ontology specific to the domain, which serves as a basis for comparing the user’s queries with the website’s content. The user’s input query is matched with the ontology and forwarded to the search engine module. This phase generates a collection of documents that are processed by the crawling system to verify the accuracy of the web pages. Only acceptable and high-quality web pages are parsed, and their content is compared with the ontology. If there is a match, the page is indexed; otherwise, it is disregarded.

The concept of gathering data from textile websites using web crawling and web scraping techniques is introduced by Muehlethaler & Albert (2021). The objective is to gather data from the websites of major retailers, such as clothing kinds, fibers, and colors. This approach of gathering data makes it easier to spot emerging trends, comprehend consumer wants, and examine consumption trends for various types of clothes. The authors create a web crawler and web scraper built on Python that is intended just for collecting information from store websites serving the textiles sector.

A learning-based focused web crawler algorithm is presented by Kumar & Aggarwal (2021), to increase the effectiveness of web crawlers. The method incorporates a modified crawler policy that helps the web crawler operate more effectively. Based on how often they are changed, the URLs are divided into static, frequent, and often modified categories. Using a k-nearest neighbors (KNN) classifier, it is possible to swiftly separate the relevant and irrelevant URLs. To further boost the effectiveness of the web crawler, the program also suggests ontology-based searching and semantic learning of keywords.

The idea of gathering unstructured data from multiple e-commerce websites is introduced by Thomas & Mathur (2019). The goal is to gather product information to provide ratings and outcomes based on this data. For web crawling, the Scrapy framework is used, enabling effective and organized data extraction from the targeted e-commerce websites. The goal of the study is to provide meaningful ratings and findings by gathering and analyzing unstructured data in order to offer insightful information on product specifics.

The Spacy and BERT machine learning frameworks are suggested by Chantrapornchai & Tunsakul (2021), to determine a hotel’s location and accommodations for travelers. Through the use of crawlers, the project gathers data on lodging, dining, shopping, and travel. The gathered data is organized and structured using ontology. To evaluate and compare the data, the machine learning frameworks Spacy and BERT are used. These frameworks make it possible to analyze and handle the information gathered, which helps the tourist sector choose appropriate areas and lodging options.

A framework is presented by Alexandrescu (2018), which is used to gather information from various online stores, particularly those that offer board and card games. The framework is developed by using Java as a programming language, and MongoDB as a database. The Google Cloud Platform, which provides scalability and reliability for data collection and archiving, is also used to build up the framework. Using board and card games as its primary domain, the framework is designed to crawl and retrieve pertinent data from various online retailer’s websites. Various online scraping techniques are used to extract information regarding product specifications, costs, user reviews, and other relevant data from different websites. For further processing and analysis, the retrieved data is subsequently kept in a MongoDB database. By utilizing Java and MongoDB the system is deployed on the Google Cloud Platform, the framework provides a dependable and scalable solution for obtaining and managing data from various shopping websites related to board and card games.

An automated method for creating datasets is introduced by Jiménez-Ruiz et al. (2020), which matches tabular data with knowledge graphs. Classes, instances, and their characteristics are extracted using this approach. Then, using SPARQL, a set of queries is created for each class. A refining phase is carried out to improve the correctness of the tables for matching purposes before the produced queries or tables are displayed in a tabular style. A smaller group of tables that are more difficult to match are then gathered.

The work in Selvalakshmi & Subramaniam (2018), describes the development of a novel semantic information retrieval system that uses feature selection and classification to improve similarity scores, along with a semantic information retrieval system based on ontology and Latent Dirichlet Allocation (LDA).

A semantic, ontology-enabled search model is discussed by Kyriakakis et al. (2019), which specifically focuses on the mapping and semantic annotation of bioinformatics resources. This search approach utilizes advanced ontologies like EDAM and SWO. Apache SOLR free-text annotator is employed for text parsing and term identification. Additionally, the model includes a speech recognition feature that enables search results based on spoken queries. The performance of the model is evaluated using the SEQanswers bioinformatics forum.

Another novel approach presented recently in Liu et al. (2023) proposes a novel focused crawler that addresses traditional web crawler challenges by integrating an enhanced tabu search algorithm with domain ontology and host information memory. The domain ontology, developed using the formal concept analysis (FCA) method for topic description, guides the crawler’s navigation. An improved tabu search (ITS) algorithm, along with host information memory, is employed to select the next hyperlink strategically. Additionally, a comprehensive priority evaluation method is designed to assess unvisited hyperlinks effectively, mitigating the issue of topic drifting. Experimental results in the tourism and rainstorm disaster domains demonstrate that FCITS_OH surpasses other focused crawling algorithms, enabling the collection of a greater quantity and higher quality of webpages.

Information extraction

OntoHuman is a toolchain designed to engage humans in the process of automatic information extraction and ontology enhancement (Opasjumruskit et al., 2022). It utilizes Ontology-Based Information Extraction (OBIE) to extract information in the format of key-value-unit tuples from PDF documents, guided by ontologies. The application of OntoHuman is versatile and applicable to documents across various engineering domains, offering an intuitive and collaborative approach to working with ontologies for users. The toolchain includes the Document Semantic Annotation Tool (DSAT) and continuously trained ontology (ConTrOn), a standalone application with a web API, which facilitates the automatic extraction of information from PDF documents based on ontologies.

An intelligent web data extraction system, created exclusively for the e-commerce industry, is suggested by Selvy et al. (2022). The data extraction is done by combining the You Only Look Once (YOLO), long short term memory (LSTM), and residual neural network (ResNet) models. The relevant data is extracted from e-commerce-related websites using these deep learning models. For object detection and identification the YOLO is adapted, the successive sequence modeling and analysis are done by using LSTM. Feature extraction from the images is performed through ResNet. The proposed solution intends to provide more reliable and effective data that is to be extracted from e-commerce websites by including these models, making it simpler to collect pertinent data for a number of e-commerce-related applications.

Teng, Day & Chiu (2022) introduce the Chinese financial information extraction system (CFIES) designed for constructing a financial knowledge graph (FinKG). This system is employed to recognize, dissect, forecast, or assess a company’s prospective financial worth using financial documents. The pertinent information is extracted from these financial documents and subsequently incorporated into the financial knowledge graph, which helps in finding the financial information and making better financial decisions.

An open-source framework called Knowledge Graph Exploration and Visualization (KGEV), leverages the Human Phenotype Ontology (HPO) to connect genes with clinical phenotypes and disease names (Peng et al., 2022). Utilizing neo4j as the graph database, the framework enables interactive exploration of knowledge graphs (KG), allowing users to search for specific entities and relationships across multiple data sources. The framework was applied to COVID-19 as a use case, demonstrating how the integration of data from diverse sources reveals various biomedical relationships, including genes, proteins, phenotypes, drugs, and diseases. The developed web application facilitates answering simple questions related to COVID-19.

One of the recent works in this area is presented by Chen et al. (2023) that introduces the article related to the creation of a knowledge graph known as the critical infrastructures protection knowledge graph (CIPKG). Its primary aim is to enhance defense mechanisms against emerging threats and bridge existing information gaps. This endeavor involves the development of two distinct ontologies: one centered around attack patterns, vulnerabilities, and threats for offensive purposes, and the other geared towards defense strategies. To ensure precise and effective information extraction, a combination of the BiLSTM and CRF model, in conjunction with the BERT model, is employed.

A framework called ANTON was presented in Hosseinkhani, Taherdoost & Keikhaee (2019), which leverages crime ontology to analyze cybercrime cases and assist in crime investigations using the ANT-based focused crawling technique. ANTON utilizes support vector machines (SVMs) and naive Bayesian AI methods for crime classification. The system consists of two key components: the crime ontology builder and the focused crawler. To address the match-making challenges in ant-based analysis, the ANTON methodology incorporates ontology-based analysis.

Knowledge-Enhanced Graph Inference (KEGI) is one of the recent works in Han & Wang (2024), that aimed at extracting knowledge from extensive text paragraphs. The solution introduces a framework for constructing knowledge graphs tailored to the industrial domain. This framework relies on knowledge-enhanced document-level entity and relation extraction, leveraging ontology and a bidirectional long short-term memory conditional random field (BiLSTM-CRF) framework. We utilize the SPFRDoc and SPOnto datasets to validate our approach. This solution exhibits promise for addressing fault diagnosis and facilitating intelligent question answering in real-world production scenarios.

ONTBOT, an ontology-based chatbot is presented by Vegesna, Jain & Porwal (2018), which is created exclusively for the e-commerce industry. The Python programming language is used to create the chatbot, which uses ontology-based methods for information extraction. The chatbot can recognize and reply to customer inquiries about e-commerce by utilizing ontology and extracting pertinent data to give precise and beneficial answers. Through natural language communication with the chatbot, ONTBOT aims to improve the user experience while seeking information and support in the e-commerce space.

The recent trends toward Question-Answering systems for the financial domain have seen the usage of the Large Language Model (LLM). Examples include Islam et al. (2023), Zhao et al. (2024), Srivastava, Malik & Ganu (2024) and Sarmah et al. (2023). However, initial results show that LLMs are still not very successful in capturing factual information and various semantic relations. Hence the results are not expected to be of high accuracy as shown by Sarmah et al. (2023) where, the semantic similarity, even after augmenting the LLMs with retrieval component, was not achieved above 60%.

Discussion

The aforementioned studies provide an overview of researchers’ efforts in proposing and constructing web crawlers for information gathering and extraction purposes. In general, ontology-based information extraction systems are predominantly developed to achieve domain-specific information extraction, with the aim of enhancing relevance. These domains encompass a wide range of areas such as education/e-learning, cybercrime, construction, garments websites, healthcare, and tourism. The listings are placed in Table 1, that shows a gap in the domain of finance as it does not gain much attention. Moreover, very few works have provided an initial discussion on financial knowledge graphs. As per the available literature, this work is the first attempt towards automated information extraction from an annual financial report using ontology and modeling the facts as graphs using knowledge graphs.

Table 1 Summary table.

References	Year	Task	Technique	Domain	
Ibrahim & Yang (2019)	2019	Information gathering using a web crawler based on ontology (WCO)	Ontology
Protégé
Java
MySQL	Education	
Sowunmi et al. (2018)	2018	Information retrieval of educational content	Semantic web based framework based on ontology and semantic web search engine /crawler.	Education/eLearning	
Kyriakakis et al. (2019)	2019	Semantic, ontology-enabled search model	ontologies Apache
Solr free-text annotator.	Bio-informatics	
Wei & Liu (2019)	2019	Knowledge graph creation of the Hundred Schools of
Thought is proposed.	Ontology
web crawler Neo4j,
Knowledge graph.	Generic	
Jiménez-Ruiz et al. (2020)	2020	Knowledge graph
matching, an automated technique for information extraction	SPARQL
Knowledge graph	Generic	
Chen et al. (2023)	2023	A management knowledge graph approach for critical infrastructure protection: Ontology design, information extraction and relation prediction	Ontology
Knowledge graph
BiLSTM
CRF model
BERT model	Critical infrastructure protection	
Hong, Lee & Yu (2019)	2019	Ontological approach for automated information gathering	Web crawler
Ontology
.	Construction	
Muehlethaler & Albert (2021)	2021	Information retrieval using web crawling and web scraping tools	Web crawler web scraper is developed for collecting data.	Garments	
Thomas & Mathur (2019)	2019	Collecting unstructured data from various e-commerce websites	Scrapy
framework for
web crawling	e-commerce	
Vegesna, Jain & Porwal (2018)	2018	ONTBOT, an ontology-based chatbot, for information extraction	Ontology-based chatbot using python	e-commerce	
Alexandrescu (2018)	2018	Information gathering from different shopping
websites regarding board and card games.	Java, MongoDB	e-commerce	
Selvalakshmi & Subramaniam (2018)	2018	Semantic information retrieval system	Feature selection algorithm and an intelligent ontology and Latent Dirichlet Allocation based semantic information retrieval algorithm	Generic	
Hosseinkhani, Taherdoost & Keikhaee (2019)	2019	ANTON framework is designed which utilize crime ontology in order
to analyze the
cyber-crime cases and helps in crime investigations	Ant-colony algorithm
Ontology
SVM
Naive Bayesian	Crime	
Kumar & Aggarwal (2021)	2021	To enhance the efficiency of web crawlers, and to increase the web crawler’s performance.	Ontological based searching, Semantic learning of Keywords
KNN classifier	Generic	
Chantrapornchai & Tunsakul (2021)	2021	Spacy and BERT machine learning to
know the location and
hotel accommodation for the tourism industry.	web-crawlers ontology, Spacy, and BERT are used for data evaluation and comparison	Tourism	
Selvy et al. (2022)	2022	Information extraction system for e-commerce domain	YOLO, LSTM & Residual Neural Network (ResNet) for data extraction	e-commerce	
Teng, Day & Chiu (2022)	2022	Text mining with information extraction for Chinese financial knowledge graph	Text mining
Financial knowledge graph	Finance	
Opasjumruskit et al. (2022)	2022	OntoHuman: Ontology-Based Information Extraction Tools with Human-in-the-Loop Interaction.	Ontology-Based Information Extraction (OBIE),
Document Semantic Annotation Tool (DSAT), Continuously trained ontology (ConTrOn)	Engineering domain	
Peng et al. (2022)	2022	Expediting knowledge acquisition by a web framework for Knowledge Graph Exploration and Visualization (KGEV): case studies on COVID-19 and Human Phenotype Ontology.	Ontology, knowledge graph	Disease/Health	
Liu et al. (2023)	2023	A new focused crawler using an improved tabu search algorithm
incorporating ontology and host information	Focused web crawler, Ontology, Tabu search algorithm	Tourism and rainstorm disaster domains	
Han & Wang (2024)	2024	Knowledge enhanced graph inference network based entity-relation extraction and knowledge graph construction for industrial domain	Ontology,
Knowledge Graph, BiLSTM-CRF, knowledgeenhanced graph inference (KEGI)	Industrial domain	

This work has the following contributions: Financial ontology-based information extraction

Modeling relevant information as a financial knowledge graph (FKG).

Analyzing financial institution’s performance in an automated way.

An ontological representation has been employed to model the domain of Finance to extract relevant semantic information. This leads to the achievement of high recall and precision that will be discussed in the result section.

The financial knowledge graph will help in answering the semantic query from the annual financial reports of the banks. The process is discussed in the next section.

A financial knowledge graph is modeled for each bank. The integrated view of different banks will help in analyzing the banks’s performance through SPARQL-based graph queries.

Materials and Methods

The flow of the work is shown in Fig. 3. It shows the major steps involved in this study. The ontology-based information extraction is applied to the textual annual financial reports, the extracted data will be inserted in the knowledge graph and the queries are executed, which shows the results of the competency questions.

Figure 3 Major steps involved in this research.

As stated in Fig. 3, this work is divided into the following parts: Retrieval of annual financial report

Extending ontology and designing competency questions

Information extraction

Information representation using knowledge graph

Answering intelligent queries

Annual financial report retrieval

A Python-based focused web crawler is developed, which is dedicated to a particular target as specified in the query, used for retrieving and downloading the annual financial reports as per the defined target. Unlike conventional crawlers, focused crawlers do not crawl the whole web. Focused crawlers are primarily responsible for crawling only the areas of the web that are relevant to the specific topic (Khan & Sharma, 2016).

The focused crawler is developed using Python libraries including Requests, BeautifulSoup, and PdfFileReader. Python’s ‘re’ module has been used for regular expressions to retrieve the focused information from web pages. The system flow is as under: (i) Crawler initiates a request to the specified URL. (ii) The request ‘gets’ the HTML content. (iii) The crawler parses the content using the BeautifulSoup library. (iv) The parsed contents are evaluated using ‘re’ module for regular expressions to locate annual financial reports on the page. (v) The matched links are then extracted and added to a list.

The determination of the focus is one of the challenges as compared to the traditional web crawlers.This has been achieved by selectively retrieving only the links to annual financial reports available on the provided web page. This selective approach ensures that only relevant PDF files, typically containing annual financial reports, are captured, while disregarding unnecessary files and links. This efficiency saves time by eliminating the need to sift through irrelevant material, thereby streamlining the data retrieval process for further analysis and information extraction purposes.

Extending ontology and designing competency questions

The ontology in Zehra et al. (2021) is extended to model the domain of annual financial reports. The data pertaining to the vocabularies, entities, and terminologies associated with annual financial reports across diverse commercial banks is gathered. This collection aids in the formulation of properties and relationships among these elements. Through analysis and incorporation of these terminologies, classes and their relationships are established. Subsequently, these classes are organized into a taxonomic hierarchy, following the methodology outlined in Noy & McGuinness (2001). The model has been tested on various commercial banks including (MCB (mcb.com.pk), ABL (abl.com), HBL (hbl.com), UBL (ubldigital.com), ASKARI (askaribank.com), BAFL (bankalfalah.com), BAHL (bankalhabib.com), BISLAMI (bankislami.com.pk), FBL (faysalbank.com), MBL (meezanbank.com), NBP (nbp.com.pk) and SCB (sc.com/pk)).

The ontology is validated through the W3C ontology validator and the validation results are available online (Mohsin, 2024). The ontologist needs to determine the purpose of ontology development (Noy & McGuinness, 2001). For this purpose, the competency questions need to be determined by reviewing the various annual financial reports of the banks and their usage by the stakeholders. These questions are designed by keeping in mind the queries of the layman and investors related to the investment opportunities and the financial benefits offered by the different banks and financial institutions in their annual financial statements, also the design process of the competency questions involves a thorough examination and analysis of key factors associated with annual financial statements, drawing insights from information accessible on diverse financial websites (Investopedia, 2021).

The competency questions are mentioned below:

I. Competency questions

Q1. Does the bank offer an Internet banking facility?

Q2. Does the bank offer a mobile banking facility?

Q3. Which banks offer Internet banking facilities?

Q4. Which banks offer mobile banking?

Q5. Which banks announce dividend in the year 2020?

Q6. What is the cash dividend per share of the banks in 2020?

Q7. What are the earnings per share after tax of a particular bank in the year 2020?

Q8. What is the interest income earned by the bank in 2020 in billions?

Q9. What is the gross markup income of the bank in the year 2020 in billions?

Q10. What is the total number of accounts of different banks in the year 2020?

Q11. What is the total number of branches of different banks in the year 2020?

Q12. What is the total number of employees of different banks in the year 2020?

Q13. What is the total number of ATMs of different banks in the year 2020?

Q14. What is the total number of Internet banking customers of the bank in the year 2020?

Q15. What is the total number of mobile banking customers of different banks in the year 2020?

Q16. What is the total number of credit card customers of different banks in the year 2020?

Q17. What is the total number of Internet banking transactions of the bank in the year 2020?

Q18. What is the total number of mobile banking transactions of the bank in the year 2020?

Q19. What is the total volume of Internet banking transactions of the bank in the year 2020 in millions?

Q20. What is the total volume of mobile banking transactions of the bank in the year 2020 in millions?

The competency questions help in determining the domain. This leads to the generation of classes/subclasses, relationships and instances. Figures 4–6 show the ontology developed for information extraction. The figures show different classes, instances and their relationships. This figure models the facts as rules in the domain of financial reports using ontological representation. These facts help in extracting relevant details from the reports.

Figure 4 View of the customer class of the ontology.

Figure 5 View of the dividend, employee, trade and transaction classes of the ontology.

Figure 6 View of the different classes of the ontology.

The following subsection details the information extraction from the financial reports. The system has been implemented on the physically available financial reports based on their complexities.

Due to the lack of common semantics, it has been found that each bank is using different terms for the same concept. The ontological representation helps in resolving the semantic differences using ‘same as’ relationships or equivalent classes. Examples of such are shown in Table 2.

Table 2 Semantic differences.

Sub class name	Equivalent to	
No._of_credit_cards_customers	No._of_customers_Credit_Cards	
No._of_debit_cards_customers	No._of_customers_Debit_Cards	
No._of_Internet_Banking_subscribers	No._of_customers_Internet_Banking	
No._of_Mobile_Banking_subscribers	No._of_customers_Mobile_Banking	
Earnings_per_share	Earnings_per_share_(after_tax)

EPS

	
Interest_earned	Interest_Income_Earned	
Profit_After_Tax	Profit_After_Taxation

PAT

	
Auto_Loan	Car_Loan	

Figure 7 shows the view of the ontology using the Protégé (protege.stanford.edu) panel. It shows different classes and their subclasses to supervise the process of information extraction. The ontology guides the information extraction phase in extracting relevant data. The OWL code and the view of the complete ontology are available online (Supplemental files).

Figure 7 View of the ontology using Protégé.

Information extraction

The information extraction phase provides automation in the retrieval of annual financial reports from the online repository and extracting relevant data from the financial reports. A web crawler is developed in Python to search the annual reports from the repository which is processed with a Python programming script to retrieve the financial statements of the bank available in unstructured data format.

In the second task, the retrieved annual report (MCB Bank Limited, 2022) in Fig. 8 is transformed into a tabular format in an automated way. The data of the annual financial report 2020 of the bank (MCB Bank Limited, 2022) is converted into the W3C compliant CSV format file. The CSV file contains many relevant and non-relevant items. The relevant terminologies based on rules are retrieved using the ontology developed (Figs. 4–6). The relevant data is extracted from the annual financial report 2020 of the bank with the proposed ontology.

Figure 8 View of the annual financial statement of the MCB bank (Source: https://www.mcb.com.pk/assets/documents/Annual-Report-2020.pdf).

The different steps involved in the entire process are as under:

Step 1: The retrieval of the annual financial report from the online repository.

Step 2: Conversion of textual data into W3C compliant tabular data format

Step 3: Extraction of relevant data from the tabular data

Step 4: Modeling relevant data into the knowledge base using a financial knowledge graph

Step 5: Visualization for querying

Figure 9 shows the importing and querying view of the knowledge graph (using Neo4J software), of the bank for the year 2020. The answers retrieved to the queries are shown in the result section.

Figure 9 Neo4j table & text view, importing and querying the data of the annual financial report of the MCB bank 2020 (MCB Bank Limited, 2022).

Financial knowledge graph

In this research, the knowledge graph is used as a data repository to store and query the results related to the different queries of the investors regarding the financial stability and market position of the bank, which includes the profit/income of the bank, the different latest banking facilities provided by the bank including online and mobile banking. The volume of online transactions can be used as a strong indicator, which shows the customer’s satisfaction and their confidence and trust regarding the online services of the particular bank. The knowledge graph serves as a structured representation of the bank’s data, enabling efficient storage, retrieval, and analysis of information.

It allows investors to gain insights into the bank’s financial performance, assess its competitive position in the market, and make informed decisions based on the available data. The information extracted from the annual financial reports of the financial institutes is modeled in the knowledge graph for querying and answering the competency questions. For storing the extracted data this article utilizes Neo4J software as a graph database, used for importing the extracted data, which is in the CSV data format, into a knowledge graph, the extracted data contains the terminologies and values of the annual financial report. The detailed knowledge graph is shown in Fig. 10.

Figure 10 Knowledge graph visualization.

The developed knowledge graph helps in answering intelligent queries beneficial in solving use cases presented in the initial subsection. It can provide relevant and reliable information to various information retrieval systems including question answering, recommender, and search engine systems. The next subsection details the performance of the knowledge graph developed by showing the results of implementation.

Results

Figure 11 shows the selected results of querying financial information. The snapshots show the query results related to the cash dividend (competency question number 5), cash dividend per share (competency question number 6), and dividend payout ratio of the banks for the year 2020.

Figure 11 Neo4j table data view, presents the cash dividend related data of the bank/financial institutions for the year 2020.

Figure 12 shows the snapshots of query results of the information regarding the number of branches (competency question number 11), number of accounts (competency question number 10), number of employees (competency question number 12), number of ATMS for the year 2020 (competency question number 13).

Figure 12 Neo4j table data view, containing the information regarding number of branches, number of accounts, number of employees and number of ATMS of MCB bank/financial institutions for the year 2020.

Figure 13 contains the information regarding the number of Internet banking customers (competency question number 14), number of mobile banking customers (competency question number 15), and number of credit card customers (competency question number 16), for the year 2020. As previously stated, analyzing these outcomes leads to making inferences about the financial prospects of the institutions.

Figure 13 The Neo4j table data view contains the information regarding number of internet banking customers, number of mobile banking customers, and number of credit card customers of the MCB bank for the year 2020.

The summarized results of all the competency questions related to the Annual Financial Report of the MCB bank answered for the year 2020 are shown in Table 3.

Table 3 Summarized results of the competency questions.

Competency question number	Query results	
Q1	Yes	
Q2	Yes	
Q3	MCB	
Q4	MCB	
Q5	MCB	
Q6	20	
Q7	24.5	
Q8	145,772,451	
Q9	75,843,439	
Q10	8,217,065	
Q11	1,429	
Q12	13,643	
Q13	1,434	
Q14	198,939	
Q15	1,396,475	
Q16	84,542	
Q17	691,553	
Q18	2,793,156	
Q19	29,200	
Q20	78,674	

No alternate system is available to answer the competency questions mentioned earlier. In general, investors use search engines or Q-A-based systems for the needful, however, due to the non-availability of data in the knowledge base their performance is not as desired. The output of our system as a knowledge graph may however be used by existing search engines/recommender/Q-A systems as knowledge graph embedding’s to provide meaningful results. The existing results of the answers to the competency questions taken from the annual financial reports of the different banks are shown in Table 4.

Table 4 Answers to the competency questions with precision and recall values.

Bank	Q1	Q2	Q3	Q4	Q5	Q6	Q7	Q8	Q9	Q10	Q11	Q12	Q13	Q14	Q15	Q16	Q17	Q18	Q19	Q20	Available	Not available	Precision	Recall	
MCB	Y	Y	Y	Y	Y	Y	Y	Y	Y	Y	Y	Y	Y	Y	Y	Y	Y	Y	Y	Y	20	0	1	1	
ABL	Y	Y	Y	Y	Y	Y	Y	Y	N	N	Y	Y	Y	N	N	N	N	N	Y	Y	13	7	1	0.65	
ASKARI	Y	Y	Y	Y	Y	Y	Y	Y	N	N	Y	Y	Y	N	N	N	N	N	N	N	11	9	1	0.55	
BAFL	Y	Y	Y	Y	Y	Y	Y	Y	Y	N	Y	Y	Y	N	N	N	N	N	N	N	12	8	1	0.6	
BAHL	Y	Y	Y	Y	Y	Y	Y	Y	Y	N	Y	N	Y	N	N	N	N	N	N	N	11	9	1	0.55	
BISLAMIC	Y	Y	Y	Y	N	N	Y	N	Y	Y	Y	Y	Y	Y	Y	Y	N	N	N	N	13	7	1	0.65	
FBL	Y	Y	Y	Y	N	N	Y	N	N	N	Y	N	Y	N	N	N	N	N	N	N	7	13	1	0.35	
HBL	Y	Y	Y	Y	Y	Y	Y	Y	Y	Y	Y	Y	Y	Y	Y	Y	N	Y	N	Y	18	2	1	0.9	
MBL	Y	Y	Y	Y	Y	Y	Y	Y	Y	N	Y	Y	Y	N	N	N	N	N	N	N	12	8	1	0.6	
NBP	Y	Y	Y	Y	Y	Y	Y	Y	Y	Y	Y	Y	Y	N	N	N	N	N	N	N	13	7	1	0.65	
SCB	Y	Y	Y	Y	Y	Y	Y	Y	Y	N	Y	N	Y	N	N	N	N	N	N	N	11	9	1	0.55	
UBL	N	N	N	N	Y	Y	Y	Y	Y	Y	Y	Y	Y	N	N	N	N	N	N	N	9	11	1	0.45	

Table 4 shows the answers to the competency questions with respect to the annual financial reports of different banks having operations worldwide. The dataset has been generated by applying a web crawler to gather the annual financial reports. It also shows the results in the form of high precision and recall values. The use of our financial knowledge graph as knowledge graph embedding’s will significantly improve the results.

Table 4 also satisfactorily addresses all the three use cases pertaining to profit after tax / net interest earned, and dividend per share values, which are already discussed above in detail. The higher number of available information in Table 4 with the exception of a couple of banks shows the applicability of our developed ontology by various banks which is an initial attempt towards standardization. The higher values are the result of integrating synonyms of various terminologies by banks for common concepts in ontology. The integration is now pervasive and implicit in the knowledge graph.

Figure 14 shows the graph between the numbers of correct answers to the competency questions of the different banks whose reports are publicly available and have operations in different countries in the Asian region. The figure shows that the MCB bank got the maximum number of points by fulfilling the queries of all the competency questions, while the HBL bank remains in second position by fulfilling the answers to the 18 competency questions out of 20. The results generated from our proposed solution as a financial knowledge graph, are more reliable because the data is extracted from the official source, thus leading to high precision and high recall by modeling ontological representation into knowledge graph as shown in Fig. 10 and querying through it as shown in Figs. 12 and 13.

Figure 14 Graph between number of correct answers to the competency questions of the different banks of Pakistan.

Conclusions

This study presents an approach for extracting relevant data from the annual financial reports of financial institutions. As a proof of concept, the approach is applied to publicly available annual reports for the year 2020. The task is challenging due to the lengthy and complex nature of these reports, which are only available in textual format, making the data vulnerable to machine readability. To overcome this challenge, we successfully converted the representation system from a text-based PDF to a W3C-compliant CSV-based representation system. Other available structures, such as JSON and XML, can also be utilized for future integrations with existing information systems. The choice of CSV format in this work was driven by the primary objective of this work “to transform the textual format into a knowledge graph”. This was achieved by developing an ontology that matches the extracted terminologies with the concepts in the finance domain. The ontology serves as a common ‘platform’ bridging the gap between textual and graphical representations. The extracted data is then transferred into the knowledge graph, from which the results are generated to answer our competency questions.

The transformation of raw financial data into ontology-based representation enables effective querying of financial report data, that ensures relevant and reliable results gathered from different banks through their publicly available reports. Although the ontology may not model all concepts for every bank, it can be easily extended to incorporate additional terminologies and concepts as desired by individual institutions. This has already been achieved and demonstrated in other domains, such as healthcare and academia.

In future extensions of this work, the generated knowledge graph can be leveraged for natural language generation, enabling visualization of the results through various means such as chatbots, storytelling, recommender systems and Q-A systems. We are currently developing a storytelling-based system that will provide a summarized overview of the financial reports for investors using this knowledge graph.

Supplemental Information

Supplemental Information 1 Financial Ontology Visualization.

Supplemental Information 2 Information Extracted from the Annual Financial Report of the MCB Bank for the year 2020.

Supplemental Information 3 MCB Annual Financial Report for the Year 2020.

Supplemental Information 4 Financial Ontology Code with w3c's (.owl) extension.

Additional Information and Declarations

Competing Interests

Author Contributions

Data Availability

The authors declare that they have no competing interests.

Syed Farhan Mohsin conceived and designed the experiments, performed the experiments, performed the computation work, prepared figures and/or tables, and approved the final draft.

Syed Imran Jami conceived and designed the experiments, performed the experiments, analyzed the data, authored or reviewed drafts of the article, and approved the final draft.

Shaukat Wasi analyzed the data, authored or reviewed drafts of the article, and approved the final draft.

Muhammad Shoaib Siddiqui performed the computation work, prepared figures and/or tables, performed Language Editing work on final manuscript, and approved the final draft.

The following information was supplied regarding data availability:

The code is available in the Supplemental File.

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
