# Peer review of "An automated information extraction system from the knowledge graph based annual financial reports"

_PeerJ Computer Science, doi:10.7717/peerj-cs.2004_

## Round 0.1 · original submission · Major Revisions

· Academic Editor

Major Revisions

The authors should revise the article based on the reviewer comments and also provide detailed responses.

**Language Note:** The review process has identified that the English language must be improved. PeerJ can provide language editing services - please contact us at [email protected] for pricing (be sure to provide your manuscript number and title). Alternatively, you should make your own arrangements to improve the language quality and provide details in your response letter. – PeerJ Staff

·

Basic reporting

The paper presents a semantic web-based solution to extract information automatically from annual financial reports of banks and financial institutions. This is an interesting research work, though the paper is required to provide a more clarity in the methodology and results discussions. I have identified specific areas for improvement

The abstract and introduction could be more concise, clearly outlining the paper's aims, methodology, and findings

Figures needs to be more clear, such as the text in figure 4 is pixelated / blurry after zooming.

Experimental design

This work has the following contributions:
1. Financial ontology-based information extraction
2. Modeling relevant information as a financial knowledge graph (FKG).
3. Analyzing financial institution's performance in an automated way.
It is advised to briefly explain how these contributions has been validated

Expand the review to include a broader range of relevant studies for 2022 and 2023, showing how this work builds upon or differs from existing research.

Validity of the findings

Provide more in-depth explanations of the methodologies, particularly the ontology design and knowledge graph implementation, including the results validation.

Figure 4 in the paper shows an ontology developed for information extraction from financial reports. This ontology includes various classes, instances, and their relationships, crucial for extracting relevant details from the reports. How does the ontology in Figure 4 handle the variations in terminologies across different financial reports, and how does it ensure accuracy in data extraction and representation?

Additional comments

The paper uses competency questions to test the ontology and the information extraction system. A more detailed description of how these questions were developed and how they align with the goals of the research would enhance the validation process.

Table 1 only includes only one paper from 2023, however, Recent research in the field of automated information extraction from financial reports has shown a variety of approaches and methodologies. One significant approach is the use of large language models (LLMs) to extract information from financial reports [arXiv preprint arXiv:2310.10760.]. I will advise to improve this table, and add some relevant recent studies.

Reviewer 2 ·

Basic reporting

Briefly describe the semantic web-based solution and ontological approach employed.
Present the results obtained, with a focus on answering competency questions and evaluating precision and recall measures.
Write the significance of the proposed solution in addressing challenges related to unstructured financial data in abstract conclusion.
Include relevant keywords that capture the essence of the research, such as semantic web, knowledge graph, financial reports, ontology, information extraction.
The background is not cleared. Moreover, what is the need of this section. Authors should reformulate this section to clear the formatting and readability.
Figure 4 is not clear in which you add different classes of the Ontology.
How did the author quantify this study? There is a suggestion need for an explicit section demonstrating the overall review methodology of the proposed article.
What is the need of use cases in this scheme? What is the need for such diagrams? Moreover, the diagrams are too low in quality.
Results are not well explained. The authors should provide a clear description of the results.
Overall, the sections of the complete article are not well coherent. Please see a well cited paper to reformulate the sections of the article.

Experimental design

Included in basic reporting.

Validity of the findings

Included in basic reporting.

·

Basic reporting

Abstract:
The abstract effectively introduces a semantic web-based solution for addressing challenges in extracting information from financial reports. It is well-structured, but providing more specifics on semantic differences, implementation details, quantitative results, and highlighting the novelty would enhance its overall clarity and impact.

Related Work

1. Evaluation and Critique: While the section provides summaries of various studies, there's limited evaluation or critique of the methodologies used in these studies. A more critical analysis of the strengths and limitations of each approach could provide valuable insights for the readers.

2. Comparison with Proposed Work: The section mentions a comparison with the proposed work but lacks a detailed and structured comparative analysis. Providing a clear and structured comparison, perhaps in a tabular format, would enhance the reader's understanding of how the proposed work differs or builds upon existing methodologies.

3. Thematic Organization: The studies are presented in a somewhat chronological order, but organizing them thematically based on similarities in methodologies or domains might improve the coherence of the presentation.

Experimental design

Materials & Methods

1. Clarity in Annual Financial Report Retrieval: While the use of a focused web crawler is mentioned, more details on how the Python-based focused web crawler is developed and its specific functionalities could enhance the clarity of this step. Providing insights into challenges or considerations in the retrieval process would add depth.

2. Competency Question Determination: The section briefly mentions the determination of competency questions, but it could benefit from a more detailed explanation of the process. How were these questions formulated? Were stakeholders involved in the determination process? Providing insights into these aspects would enhance the robustness of the methodology.

3. Linking Methodology Steps to Competency Questions: The section does not explicitly link each step of the methodology to the corresponding competency questions. It would be beneficial to explicitly mention how each part of the methodology contributes to addressing or answering the competency questions.

4. Integration of Terminologies: While the extension of the ontology is mentioned, additional details on how terminologies from various annual financial reports are integrated into the ontology would provide a more comprehensive understanding of the process.

Validity of the findings

Competency Questions:

1. Diversity of Information Sources: While the competency questions cover a broad spectrum of financial information, there could be an explicit mention of the sources of information, considering that financial data can be sourced from various reports, documents, or databases. This clarification would enhance the understanding of the data extraction process.
2. User-Centric Perspective: The competency questions are primarily focused on extracting factual information. Introducing questions that reflect user preferences or sentiments might add a user-centric perspective. For example, questions related to customer satisfaction or public sentiment toward a bank could be considered.
3. Quantifiable Metrics: Some questions seek specific numerical values, such as the total number of transactions or the volume of transactions. It would be beneficial to clarify the units or metrics used for these quantitative queries, ensuring consistency in the interpretation of results.

Additional comments

The research presented a semantic web-based solution for automatically extracting relevant information from annual financial reports of banks and financial institutions. This extracted information was presented in a query-able form through a knowledge graph, which could be used to support various systems, including search engines, recommender systems, question-answering systems, and financial storytelling.

The main challenge in understanding information from these reports was the unstructured format of the data and the variation of terminologies among different reports. The proposed solution utilized an ontological approach to solve the standardization problems of terminologies across the annual financial reports of different banks. Semantic differences were identified and resolved to filter and extract the relevant data. The extracted results were then stored in a knowledge graph to make the information queryable.

The research presented an automated information extraction system that was implemented on datasets from various banks and was presented through answers to competency questions evaluated on precision and recall measures. This solution significantly reduced the complexities involved in understanding and querying financial information from unstructured annual financial reports.

This is a good work however following are some suggestions section-wise to improve the work overall.

Abstract:
The abstract effectively introduces a semantic web-based solution for addressing challenges in extracting information from financial reports. It is well-structured, but providing more specifics on semantic differences, implementation details, quantitative results, and highlighting the novelty would enhance its overall clarity and impact.

Related Work

1. Evaluation and Critique: While the section provides summaries of various studies, there's limited evaluation or critique of the methodologies used in these studies. A more critical analysis of the strengths and limitations of each approach could provide valuable insights for the readers.

2. Comparison with Proposed Work: The section mentions a comparison with the proposed work but lacks a detailed and structured comparative analysis. Providing a clear and structured comparison, perhaps in a tabular format, would enhance the reader's understanding of how the proposed work differs or builds upon existing methodologies.



3. Thematic Organization: The studies are presented in a somewhat chronological order, but organizing them thematically based on similarities in methodologies or domains might improve the coherence of the presentation.

Materials & Methods

1. Clarity in Annual Financial Report Retrieval: While the use of a focused web crawler is mentioned, more details on how the Python-based focused web crawler is developed and its specific functionalities could enhance the clarity of this step. Providing insights into challenges or considerations in the retrieval process would add depth.



2. Competency Question Determination: The section briefly mentions the determination of competency questions, but it could benefit from a more detailed explanation of the process. How were these questions formulated? Were stakeholders involved in the determination process? Providing insights into these aspects would enhance the robustness of the methodology.



3. Linking Methodology Steps to Competency Questions: The section does not explicitly link each step of the methodology to the corresponding competency questions. It would be beneficial to explicitly mention how each part of the methodology contributes to addressing or answering the competency questions.



4. Integration of Terminologies: While the extension of the ontology is mentioned, additional details on how terminologies from various annual financial reports are integrated into the ontology would provide a more comprehensive understanding of the process.



Competency Questions:

1. Diversity of Information Sources: While the competency questions cover a broad spectrum of financial information, there could be an explicit mention of the sources of information, considering that financial data can be sourced from various reports, documents, or databases. This clarification would enhance the understanding of the data extraction process.



2. User-Centric Perspective: The competency questions are primarily focused on extracting factual information. Introducing questions that reflect user preferences or sentiments might add a user-centric perspective. For example, questions related to customer satisfaction or public sentiment toward a bank could be considered.



3. Quantifiable Metrics: Some questions seek specific numerical values, such as the total number of transactions or the volume of transactions. It would be beneficial to clarify the units or metrics used for these quantitative queries, ensuring consistency in the interpretation of results.

---

## Round 0.2 · accepted · Accept

· Academic Editor

Accept

The authors have revised the article as per reviewer comments.

·

Basic reporting

Agreed on incorporated changes

Experimental design

Agree on incorporated changes

Validity of the findings

Agreed on incorporated changes

Additional comments

Agree with incorporated changes

Reviewer 2 ·

Basic reporting

The authors have made the suggested changes

Experimental design

NA

Validity of the findings

NA

Additional comments

NA

·

Basic reporting

improved

Experimental design

improved

Validity of the findings

improved

Additional comments

improved